# Isolation and Characterization of a Novel Jumbo Phage from Leaf Litter Compost and Its Suppressive Effect on Rice Seedling Rot Diseases

**DOI:** 10.3390/v13040591

**Published:** 2021-03-31

**Authors:** Ryota Sasaki, Shuhei Miyashita, Sugihiro Ando, Kumiko Ito, Toshiyuki Fukuhara, Hideki Takahashi

**Affiliations:** 1Graduate School of Agricultural Science, Tohoku University, 468-1, Aramaki-Aza-Aoba, Sendai 980-0845, Japan; ryouta.sasaki.q7@dc.tohoku.ac.jp (R.S.); shuhei27@gmail.com (S.M.); sugihiro.ando.a2@tohoku.ac.jp (S.A.); kumiko.ito.b8@tohoku.ac.jp (K.I.); 2Department of Applied Biological Sciences and Institute of Global Innovation Research, Tokyo University of Agriculture and Technology, Fuchu, Tokyo 183-8538, Japan; fuku@cc.tuat.ac.jp

**Keywords:** biocontrol agent, broad host range, leaf litter compost, *Burkholderia*, *Ralstonia*, jumbo phage, phage therapy

## Abstract

Jumbo phages have DNA genomes larger than 200 kbp in large virions composed of an icosahedral head, tail, and other adsorption structures, and they are known to be abundant biological substances in nature. In this study, phages in leaf litter compost were screened for their potential to suppress rice seedling rot disease caused by the bacterium *Burkholderia glumae,* and a novel phage was identified in a filtrate-enriched suspension of leaf litter compost. The phage particles consisted of a rigid tailed icosahedral head and contained a DNA genome of 227,105 bp. The phage could lyse five strains of *B. glumae* and six strains of *Burkholderia plantarii*. The phage was named jumbo *Burkholderia* phage FLC6. Proteomic tree analysis revealed that phage FLC6 belongs to the same clade as two jumbo *Ralstonia* phages, namely RSF1 and RSL2, which are members of the genus *Chiangmaivirus* (family: Myoviridae; order: Caudovirales). Interestingly, FLC6 could also lyse two strains of *Ralstonia pseudosolanacearum*, the causal agent of bacterial wilt, suggesting that FLC6 has a broad host range that may make it especially advantageous as a bio-control agent for several bacterial diseases in economically important crops. The novel jumbo phage FLC6 may enable leaf litter compost to suppress several bacterial diseases and may itself be useful for controlling plant diseases in crop cultivation.

## 1. Introduction

Bacteriophages (phages) are viruses that infect bacteria and exploit the metabolic processes of the host to replicate their own genome. The host bacteria are generally lysed as a result of phage infection, thereby releasing the virions, while phages that convert to the lysogenic cycle do not lyse the host cells but instead become integrated into the host genome as prophages [1]. Phages are classified based on their morphology and nucleic acids. A major group of identified phages consists of an icosahedral head containing phage genomic DNA, a tail, and other adsorption structures, and it belongs to the order Caudovirales, which comprises nine families: Ackermannviridae, Autographiviridae, Chaseviridae, Demerecviridae, Drexlerviridae, Herelleviridae, Myoviridae, Podoviridae, and Siphoviridae [2,3,4,5]. In Caudovirales, tailed phages with DNA genomes larger than 200 kbp are classified as “jumbo phages” [6]. The larger genomes of jumbo phages enable them to encode many more proteins than is possible for phages with smaller genomes, although the functions of many jumbo phage genes remain to be elucidated. Some proteins specifically encoded in the jumbo phage genome may compensate for the host proteins required for phage multiplication in the host bacteria, thus enabling jumbo phages to have a wider host range [6]. The wider host range of jumbo phages may make them especially advantageous as biocontrol agents against multiple bacterial diseases.

Thus far, more than 200 jumbo phages have been deposited in the NCBI Genome database, 52 of which are known to infect phytopathogenic bacteria, including *Ralstonia*, *Agrobacterium*, *Xanthomonas*, *Erwinia*, *Dickeya*, *Pseudomonas syringae* pv. *actinidae*, *Bacillus pumilus*, and *Serratia* sp. [7,8,9,10,11,12,13,14,15,16,17]. *Ralstonia solanacearum* is one of the most important plant pathogens and causes bacterial wilt in a broad range of hosts, including economically important crops, such as *Solanaceae* [18]. Some jumbo *Ralstonia* phages such as RSL1, RSF1, RSL2, RP12, and RP31 have been identified and studied as potential biocontrol agents to suppress bacterial wilt [8,15,19]. The jumbo *Agrobacterium* phage Atu_ph07, which was isolated from creek water, can be used to prevent crown gall disease [7]. A jumbo *Xanthomonas* phage, XacN1, was isolated from soil samples collected from orange groves. Classified as a novel jumbo myovirus, this phage has a wide host range, including *Xanthomonas citri*, the causative agent of Asian citrus canker [16]. Interestingly, the large genome of XacN1 encodes tRNAs that target codons that are frequently used by the phage but less frequently by its host. Jumbo *Erwinia* phages such as ΦEaH1, ΦEaH2, and Ea35-70, which were obtained while screening phage populations from the soil and aerial tissues of plants, have been shown to prevent fire blight, a contagious disease affecting some fruit trees from the family *Rosaceae* [10,14,20]. The jumbo *Dickeya* phages JA11, JA13, JA29, JA33, and AD1, which were isolated from river water, also have a broad host range and are also potential biocontrol agents [9]. The jumbo *Pseudomonas* phage Psa21 has been identified as a potential biocontrol agent for suppressing the bacterial canker of kiwifruit caused by *Pseudomonas syringae* pv. *actinidae* [11]. Another jumbo *Bacillus* phage, vB_BpuM-BpSp, can infect *Bacillus pumilus*, the cause of ginger rhizome rot, and was isolated from soil surrounding affected rhizomes [17]. In addition to the major phytopathogenic bacteria mentioned above, several other bacterial pathogens cause serious damage to crop production in many countries [21].

The genus *Burkholderia* includes human pathogens, environmental bacteria, and phytopathogens. Among the phytopathogens, *Burkholderia glumae* and *Burkholderia plantarii* are the causal agents of rice seedling rot and rice seedling damping-off, respectively. They are seed-borne pathogens of rice, and once they invade rice nurseries, losses can be severe [22,23]. To our knowledge, only a few phages capable of suppressing *Burkholderia* pathogens of rice have been reported [24], but they have not been characterized. Recently, the phytopathogenic *Burkholderia* phage FLC5 containing around 32 kbp of genomic DNA was isolated from leaf litter compost used for organic farming. However, the host range of the phage was limited to members of *Burkholderia* [25]. Two genome sequences of jumbo phages that infect *Burkholderia* have been deposited in the GenBank database: BcepSauron (accession number MK552141.1) and BcepSaruman (accession number MK552140.1). However, neither has been characterized for potential application as a biocontrol agent. In this study, we isolated a novel jumbo *Burkholderia* phage from leaf litter compost. The phage has potential as a biocontrol agent to suppress rice seedling rot caused by *B*. *glumae*.

## 2. Materials and Methods

### 2.1. Bacterial Strains and Culture Conditions

The following strains used in this study were kindly supplied by the NARO GeneBank (Tsukuba, Japan): *B. glumae* (MAFF accession numbers 302552, 106619, 301169, 302417, and 302746), *B. plantarii* (MAFF accession numbers 106727, 302466, 302475, 302909, 302912, and 302936), *Ralstonia pseudosolanacearum* (MAFF accession numbers 106603, 106611, 211270, and 301485), and *Ralstonia syzygii* subsp. *indonesiensis* (MAFF accession numbers 211271 and 327032). The strains of *B. glumae* and *B. plantarii* were cultured in a potato-peptone-glucose (PPG) medium (0.5% (*w*/*v*) peptone, 0.5% (*w*/*v*) glucose, 0.3% (*w*/*v*) Na_2_HPO_4_·12H_2_O, and 0.05% (*w*/*v*) KH_2_PO_4_ dissolved in potato infusion made from 200 g of potato per 1 L of water) [26]. The strains of *R. syzygii* subsp. *indonesiensis* and *R. pseudosolanacearum* were cultured in a casamino acid-peptone-glucose (CPG) medium (1.0% (*w*/*v*) peptone, 0.1% (*w*/*v*) casamino acids, and 0.5% (*w*/*v*) glucose) [27]. All strains were incubated on 1.5% agar plates at 25 °C before preparing the bacterial suspension.

### 2.2. Phage Isolation and Culture Conditions

Leaf litter compost was prepared in the experimental field of the Graduate School of Agricultural Science, Tohoku University (38°15′ N, 140°49′ E) as follows: 135 L of leaf litter from hardwood trees and 45 L of rice bran were piled together, and 20 L of water were added. After the temperature of the heap surpassed 60 °C, the heap was turned and mixed. To maintain aerobic conditions, the mixing procedure was repeated 5 times.

Small batches of the compost (6.5 g of fresh weight) were suspended in 20 mL of the PPG medium, and the medium was shaken vigorously for 2 h at 25 °C. Coarse soil particles were removed by centrifuging at 5000× *g* for 5 min at 25 °C, followed by filtering through a mixed cellulose ester membrane (pore size: 0.45 µm). Then, 10 mL of the filtrate were mixed with 10 µL of 1 M CaCl_2_ and 1 mL of the *B. glumae* MAFF302552 suspension adjusted to 10^8^ cfu/mL. After shaking at 180 rpm overnight at 25 °C, the enriched culture was centrifuged at 5000× *g* for 5 min at 25 °C and filtered as described above. Then, 0.5 mL of the filtrate were mixed with 0.5 mL of the *B. glumae* MAFF302552 suspension (10^8^ cfu/mL) and 5 mL of the PPG top agar medium containing 0.5% agar. The mixture was immediately overlaid on 90-mm petri dishes containing 1.5% PPG agar and allowed to solidify. The experimental procedure described here is referred to as the double agar layer method. After incubating overnight at 25 °C, the plates were examined for plaques. Single plaques were picked off and mixed with a phage buffer (68.5 mM NaCl, 10 mM MgSO_4_, 1 mM CaCl_2_, and 10 mM Tris-HCl (pH 7.5)). After vortexing for 10 s (Vortex-Genie 2; Electro Scientific Industries, Portland, OR, USA), the phage suspension was diluted with a phage buffer and used in the double agar layer method to obtain a single plaque. To ensure that the isolates originated from a single plaque, the process involved three rounds of successive isolation, and the plaque from the final round was used for phage propagation.

The phage was routinely propagated by shaking in liquid culture for 16–22 h at 25 °C with *B. glumae* MAFF302552 in the PPG medium, with a multiplicity of infection of about 0.01. The culture was centrifuged at 5000× *g* for 5 min at 25 °C and filtered through a mixed cellulose ester membrane (pore size 0.45 μm) to prepare the phage suspension. The phage suspension was stored at 4 °C.

Phage host range analysis and phage titer measurements were performed by a spot test. A mixture of 0.5 mL of the *Burkholderia* sp. suspension (10^8^ cfu/mL) and 5 mL of the PPG top agar medium was poured onto plates containing 1.5% PPG agar. After the top agar had solidified, 10 µL of serially diluted phage suspension were spotted on the top agar and dried. As a negative control, only the buffer used for making the phage suspension was spotted. Plaques were observed after 20 h of incubation at 25 °C. For *Ralstonia* sp., the CPG medium was used instead of the PPG medium.

### 2.3. Transmission Electron Microscopy

The phage suspension was centrifuged at 169,800× *g* for 90 min at 4 °C using an ultracentrifuge (himac CS100FNX, Koki Holdings Co. Ltd., Tokyo, Japan) with an S50A-2521 rotor (Koki Holdings Co. Ltd.). The pellet was re-suspended in the phage buffer, and 10 µL of the concentrated phage suspension were applied to a copper grid (400 mesh) with a collodion membrane (Nisshin EM, Tokyo, Japan) and negatively stained with 2% phosphotungstic acid (pH 7.3) for 5 min. The grid was examined by transmission electron microscopy (Hitachi H-7650, Hitachi High-Tech Corporation, Tokyo, Japan).

### 2.4. Extraction of Phage Genomic DNA and Next-Generation Sequencing

The phage culture was centrifuged at 5000× *g* for 5 min at 25 °C, and the supernatant was filtered through a mixed cellulose ester membrane (pore size: 0.45 μm) to obtain a bacteria-free phage suspension. After centrifugation at 169,800× *g* for 90 min at 4 °C, the pellet was re-suspended in the phage buffer. Phage genomic DNA was extracted from the phage suspension with phenol–chloroform, following the method described by Sambrook and Russel [28]. To determine the complete genomic DNA sequence, a library was prepared with Nextera XT kit v. 2 (Illumina, San Diego, CA, USA) and sequenced with MiSeq v. 2 Reagent Kit Nano (Illumina) (2 × 150 bp) on MiSeq (Illumina), according to manufacturer’s instructions.

### 2.5. In Silico Analysis

Filtered reads were assembled with the SPAdes v. 3.11.1 genome assembler [29]. The NCBI nucleotide collection (nt) database was searched by blastn for nucleotide sequences similar to the assembled phage genome. Protein-coding sequences were predicted with GeneMarkS v. 4.28 (with the option “Phage” and “genetic code 11”) [30]. tRNA sequences and rRNA sequences were searched with tRNAscan-SE v. 2.0 [31] and RNAmmer 1.2 [32], respectively. The genome comparison data were visualized using EasyFig v. 2.2.2 [33]. Amino acid sequences of the predicted phage open reading frames (ORFs) were used as the query for blastp search using the NCBI RefSeq protein database (release 97) with 1e-5 set as the e-value threshold, and they were annotated following the homolog of jumbo *Ralstonia* phage RSF1 (NC_028899.1). A phylogenetic tree of the putative major head protein of the phage particle and other related phages was constructed based on the amino acid sequences. The tree was generated with neighbor-joining method using clustalW v. 2.1 [34], available on the DDBJ website (https://clustalw.ddbj.nig.ac.jp, accessed on 15 June 2020) with default settings. A proteomic tree was constructed with ViPTree v. 1.9 [35].

### 2.6. Assessment of Disease Suppression Activity Against Rice Seedling Rot 

Healthy and *B. glumae*-infected rice seeds (*Oryza sativa* ‘Koshihikari’) were used to assay the disease suppression activity of the phages according to a previously described method [36]. The seeds were infected with *B. glumae* by vacuum-infiltration for 5 min with a suspension of *B. glumae* MAFF302746 adjusted to 10^7^ cells/mL with water, followed by washing with distilled water 3 times and air-drying at room temperature overnight. The infected seeds were stored at 4 °C until use.

The phage suspension was adjusted to 10^7^ pfu/mL with water. Both healthy and *B. glumae*-infected seeds were soaked in the adjusted phage suspension and incubated for 24 h at 28 °C in the dark. As a control, each type of seed was also soaked in the PPG medium diluted with water at the same medium concentration as the phage suspension. After incubation, the liquid was discarded and the seeds were washed with water 3 times. The seeds were soaked in water and incubated at 28 °C for 24 h to induce germination.

The germinated seeds were sown on commercially available soil (Gousei Baido L, Sankensoiru. Co. Ltd., Iwate, Japan) that had been sterilized by autoclaving (HVE-50, Hirayama Manufacturing Corporation, Saitama, Japan) at 140 °C for 10 min and kept at 30 °C for 8 days under a 14-h light (14,000 lux)/10-h dark cycle in a growth chamber (KG-201 HL-D, Koito, Yokohama, Japan). The severity of symptoms for each individual seedling was evaluated according to a previously described method [36,37]. Briefly, for each plant, the disease severity score was evaluated on a scale of 0–3 (0: healthy; 3: completely dead) and the disease severity index was calculated as follows: ((1*A* + 2*B* + 3*C*)/3*N*) × 100, where *N* is the total number of plants and *A*, *B*, and *C,* respectively, represent the number of plants rated 1, 2, or 3 on the above scale. The Mann–Whitney U test (*p* < 0.01) was used to compare the infected seeds soaked in the phage suspension and those soaked in the diluted PPG medium.

To evaluate the disease suppression activity of the leaf litter compost, both infected and healthy seeds were soaked in water instead of in the phage suspension, with the latter serving as a control. Germinated seeds were sown either on the abovementioned commercially available soil or a mixture of the soil and leaf litter compost (3:1, *w/w*). The plants were cultivated for 9 days at 30 °C under a 14-h light (14,000 lux)/10-h dark cycle in a KG-201 HL-D growth chamber, and the seedlings were rated for disease severity as described above.

## 3. Results

### 3.1. Isolation of FLC6 from Leaf Litter Compost and TEM Observation of Particles

Clear plaques of approximately 0.5 mm in diameter (Figure 1A) formed on the top agar medium containing the filtrate-enriched suspension of leaf litter compost mixed with the *B. glumae* MAFF302552 suspension (10^8^ cfu/mL). The phage isolated by single plaque picking was named FLC6. When examined by TEM, the FLC6 particle exhibited a head-and-tail structure, with a tail typical of the family Myoviridae (Figure 1B,C). The icosahedral head of an intact FLC6 particle was about 150 nm in size with a rigid tail of about 225 nm in length (Figure 1B), and a virion with a contracted tail was also observed (Figure 1C). Therefore, based on its basic morphology, FLC6 appeared to belong to the family Myoviridae in the order Caudovirales.

### 3.2. Genomic Features and Comparison of FLC6 Genome with Other Phages

The complete genomic DNA sequence of FLC6 indicated that its genome was 227,105 bp long with guanine–cytosine content of 52.0%. The complete sequence was registered with the databases GenBank, EMBL, and DDBJ (accession number LC592711). Because the genome was longer than 200 kbp, FLC6 was classified as a jumbo phage. FLC6 has 241 predicted ORFs (ORF1–ORF241), which encode gene products gp1–gp241, respectively. Of these, 52 ORFs were encoded on the plus strand and 189 ORFs were encoded on the minus strand (Figure 2). Furthermore, 220 ORFs had ATG as the start codon, 15 had GTG, and 6 had TTG. A blastn search using the NCBI nucleotide collection yielded two entries that were homologous to the FLC6 whole-genome nucleotide sequence: jumbo *Ralstonia* phage RSL2 (88% query coverage and 90% identity) and RSF1 (79% query coverage and 85% identity). The genome of FLC6 also showed synteny with the genomes of two jumbo *Ralstonia* phages, namely RSF1 and RSL2 (Figure 2).

Among the predicted gene products (gp1–gp241) encoded on the 241 ORFs, amino acid sequences of 228 products showed the highest homology with the amino acid sequences of the protein encoded on the genome of either RSL2 or RSF1. In contrast, homologs of six proteins of FLC6 (gp61, gp89, gp101, gp128, gp129, and gp179) were found not in the proteins encoded in the genome of RSF1 or RSL2 but in gene products encoded by the genomes of other organisms. However, no proteins homologous to seven gene products of FLC6 (gp100, gp102, gp130, gp212, gp219, gp231, and gp241) were found in the database, and neither a tRNA sequence nor rRNA sequence was predicted in the FLC6 genome.

### 3.3. Gene Annotation of Predicted Proteins Encoded on FLC6 Genome

According to the function of proteins encoded in the RSF1 genome, the proteins predicted to be encoded in the FLC6 genome were annotated with their functions (Appendix A). Seven predicted ORFs encoded proteins that might be related to nucleotide metabolism (gp77, gp78, gp119, gp118, gp120, gp168, and gp198): dihydrofolate reductase (gp77 and gp78), the α subunit (gp119) and β subunit (gp118) of ribonucleotide reductase, anaerobic ribonucleotide diphosphate reductase subunit H (gp120), thymidylate synthase (gp168), and thymidylate kinase (gp198). Eight gene products were annotated as subunits of RNA polymerase: β subunits (gp39, gp50, gp123, and gp224) and β’ subunits (gp40, gp207, gp223, and gp237). Five ORFs were predicted to encode proteins related to DNA replication or recombination: RNase H (gp56), SbcC-ATPase (gp63), GIY-YIG type nuclease (gp69), DNA ligase (gp105), and DnaB helicase (gp127). Two gene products were annotated as lysis-related proteins, namely a soluble lytic murein transglycosylase (gp41) and a transglycosylase SLT domain-containing protein (gp54), both of which are known to degrade the β-1,4 bond of peptidoglycan in bacterial hosts. Nine FLC6 proteins showed similarity with RSF1 proteins: cupin superfamily protein (gp70), Fe-S oxidoreductase (gp71 and gp79), radical SAM superfamily (gp76 and gp81), 2OG-Fe(II) oxygenase (gp80, gp82), haloacid reductase-like hydrolase (gp162), and concanavalin A-like protein (gp88).

Sixteen gene products seemed to be the structural proteins of the virion (gp26, gp27, gp29, gp30, gp35, gp59, gp68, gp98, gp99, gp125, gp175, gp176, gp183, gp190, gp192, and gp193; see Appendix A); in particular, gp29, gp30, and gp125 were annotated as a putative tail sheath, a putative major virion structural protein, and a putative major head protein, respectively. Both gp175 and gp176 were encoded on the minus strand of the FLC6 genome and were homologous with the putative tail fiber protein of RSF1 (Table 1 and Appendix A). A blastp search showed that gp175 had a 76% sequence identity (100% query coverage) with the N-terminal half of the tail fiber protein of RSF1 and a 84% sequence identity (100% query coverage) with that of RSL2, whereas gp176 had a 95% sequence identity (50% query coverage) with the C-terminal half of the tail fiber protein of RSF1 and a 92% sequence identity (54% query coverage) with that of RSL2.

Proteomic tree analysis, which was based on whole-genome tblastx analysis, showed that FLC6 belongs to the same clade as RSF1 and RSL2 (Figure 3). Furthermore, the phylogenetic tree based on the amino acid sequence of the putative major head protein (gp125) also showed that FLC6, RSF1, and RSL2 belong to the same clade (Figure 4). Taken together, these findings indicate that FLC6 is a member of the genus *Chiangmaivirus* (family: Myoviridae; order: Caudovirales), as are RSL2 and RSF1.

### 3.4. Host Range Analysis

FLC6 lysed all the tested strains of *Burkholderia* (Table 2). Because jumbo phages generally have a broad host range and complete nucleotide sequences of FLC6 were highly homologous to those of jumbo *Ralstonia* phages, we examined the susceptibility of four strains of *R. pseudosolanacearum* and two strains of *R. syzygii* subsp. *indonesiensis* to FLC6. As shown in Table 2, two strains of *R. pseudosolanacearum* (MAFF106603 and MAFF106611) were lysed by FLC6. These results suggested that FLC6 is a novel jumbo with a having broad host range that includes *Burkholderia* spp. and *R. pseudosolanacearum*.

### 3.5. Disease Suppression Against Rice Seedling Rot by FLC6 Treatment

Leaf litter compost, which was the source of FLC6 in this study, has been shown to have the potential to suppress rice seedling rot disease caused by *B. glumae* [38]. The disease suppression activity of leaf litter compost was also confirmed by an assay in the present experiment. Rice seedlings grown on a mixture of leaf litter compost and soil showed a significantly lower severity of rice seedling rot than that observed in seedlings grown on soil alone (i.e., without the compost) (Appendix A), and the leaf litter compost did not affect the growth of healthy rice seedlings (Appendix A).

To ascertain whether FLC6 contributes to disease suppression by lysing *B. glumae*, *B. glumae*-infected or healthy rice seeds were treated with FLC6 and cultivated to evaluate the occurrence of rice seedling rot disease. As a control, *B. glumae*-infected or healthy rice seeds were treated with only diluted medium. As shown in Figure 5A, rice seedlings grown from healthy seeds grew equally well with and without FLC6 treatment, suggesting that FLC6 treatment did not affect the growth of the seedlings. Moreover, the disease severity index of seedlings grown from *B. glumae*-infected seeds was only 1.0 with FLC6 treatment, but it was 89 without FLC6 treatment (Figure 5B). These results suggest that FLC6 lyses *B. glumae*, thereby preventing bacterial growth in *B. glumae*-infected seeds and suppressing the occurrence of rice seedling rot disease.

## 4. Discussion

The jumbo *Burkholderia* phage FLC6 (Figure 1) was isolated from leaf litter compost, which is sometimes used in nursery soil for organic-farmed rice because it is known to suppress seedling rot. The leaf litter compost used in this study also exhibited disease suppression activity against *B. glumae* (Appendix A). Though the effect was likely due to a combination of multiple factors, the phage seemed to contribute to disease suppression by lysing multiple strains of *B. glumae* and *B. plantarii* in rice seeds in the soil for their cultivation. Otherwise, while organically farmed soils with a disease suppression effect have been shown to harbor more diverse and more robust bacterial structures than conventional commercial soils [37], polyvalent phages might contribute to maintaining a robust bacterial population, which can lead to the suppression of diseases.

The basic morphology of FLC6 and the length of its genomic DNA (>200 kbp) indicate that FLC6 is a jumbo phage. Thus far, two genome sequences of jumbo phages that infect *Burkholderia* have been deposited in the GenBank, EMBL, and DDBJ databases; however, neither had been characterized to date. To our knowledge, this is the first study to characterize a jumbo *Burkholderia* phage. The genome nucleotide sequence of jumbo *Burkholderia* phage FLC6 has a similarity with that of two jumbo *Ralstonia* phages, namely RSL2 and RSF1 (Figure 2). FLC6, RSL2, and RSF1 were placed in the same clade as a result of proteomic tree and phylogenetic tree analyses based on the putative major head protein (Figure 3 and Figure 4; Appendix A). Indeed, FLC6 could lyse phytopathogenic *Ralstonia* spp. in addition to *B. glumae* and *B. plantarii*. The host range of FLC6 to *Ralstonia* spp. was different from previously reported host range of RSF1 and RSL2 (Appendix A). Both *Burkholderia* and *Ralstonia* belong to the family Burkholderiaceae. Similar cross-genus infection with jumbo phages has been reported in other jumbo phages that can lyse phytopathogenic bacteria, such as some members of the genera *Erwinia* and *Pantoea*, which are in the family Erwiniaceae [13]. Thus, a broad host range seems to be a feature of jumbo phages.

Because rice is the primary food for billions of people worldwide, sustainable and effective measures to control rice pathogens are essential for safeguarding rice production. Rice seedling rot and rice seedling dumping-off diseases, which are caused by *B. glumae* and *B. plantarii*, respectively, result in severe damage at the nursery stage in rice farming; both are conventionally controlled by using chemical pesticides in modern intensive farming systems. However, extensive use of chemical pesticides has led to the emergence of chemical-resistant bacteria, which is a growing concern in food production and also in human health. For example, some strains of *B. glumae* have been shown to be resistant to oxolinic acid [39]. Recently, the demand for plant disease control methods that do not completely rely on chemical pesticides has become widespread. One such option is to use bacteriophages to control phytopathogenic bacteria. In the case of phytopathogenic *Burkholderia*, reported examples are limited to phages that suppress *Burkholderia* pathogens in rice [24]. The narrow host range of some phages limits their effectiveness in controlling plant pathogens, and a phage cocktail is often used to overcome this problem [40]. Novel jumbo phages with broad host ranges could be suitable for bio-control agents. Moreover, a phage cocktail in combination with other jumbo phages with broad host ranges can be used to cover a broader range of host pathogenic bacteria. Indeed, the application of a cocktail of different phages could more effectively suppress the disease incidence caused by a strain of phytopathogenic bacteria than the amendment of only one phage strain [41]. Thus far, some *Ralstonia* phages have been shown to have disease-suppressing activity against bacterial wilt of tomato [8,42,43,44]. FLC6, with its broad host range (*B. glumae*, *B. plantarii*, and *R. pseudosolanacearum*) shows considerable potential for use as a biocontrol agent against phytopathogenic bacteria in crop cultivation.

## Figures and Tables

**Figure 1 viruses-13-00591-f001:**
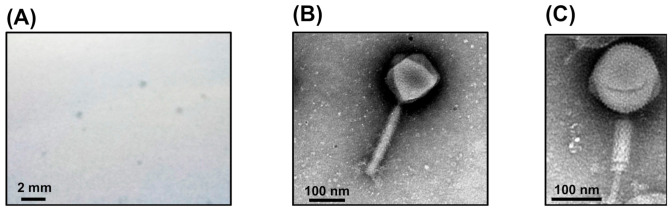
Morphology of FLC6 plaques and particles. FLC6 formed clear plaques on a lawn of *Burkholderia glumae* MAFF302552 (**A**). Both intact particles (**B**) and particles with contracted tails (**C**) were observed by TEM.

**Figure 2 viruses-13-00591-f002:**
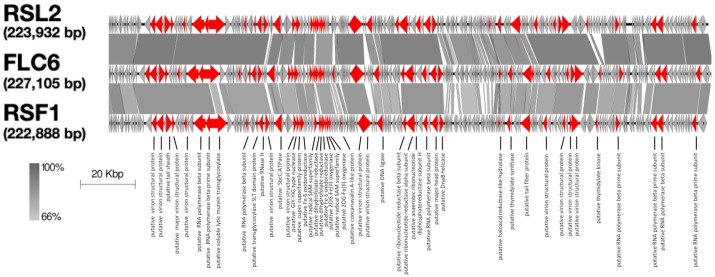
Comparison of the complete genome of FLC6 and its two related phages, jumbo *Ralstonia* phage RSL2 (RefSeq accession number NC_028950.1) and jumbo *Ralstonia* phage RSF1 (NC_028899.1). Arrows represent open reading frames (ORFs), and gray bands show similarity in nucleotide sequences. Annotated ORFs of FLC6 and the homological ORFs RSF1 and RSL2 are shown in red.

**Figure 3 viruses-13-00591-f003:**
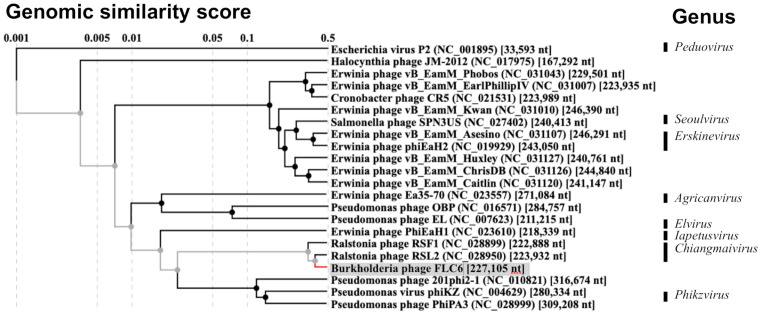
Proteomic tree of FLC6 (gray background) and other jumbo phages (*Escherichia virus* P2 was included as an outgroup). The tree was generated with ViPTree v. 1.9. The genus of phages was based on the ICTV master species list (#35). The branch to FLC6 was shown in grey and red.

**Figure 4 viruses-13-00591-f004:**
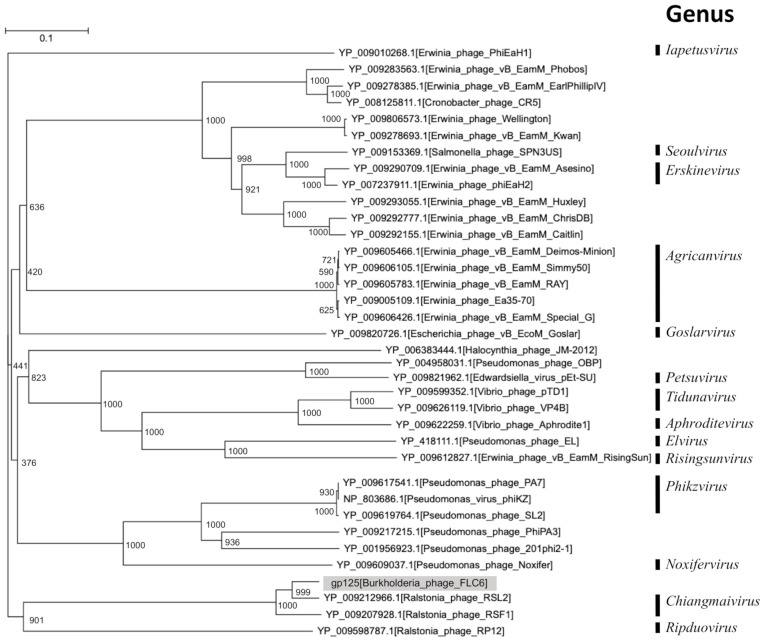
Phylogenetic tree based on amino acid sequences of major head proteins. The tree was generated with clustalW2 (on DDBJ web version with default settings) using the neighbor-joining method and a bootstrap value of 1000 iterations. Gp125 of FLC6 was shown in gray background.

**Figure 5 viruses-13-00591-f005:**
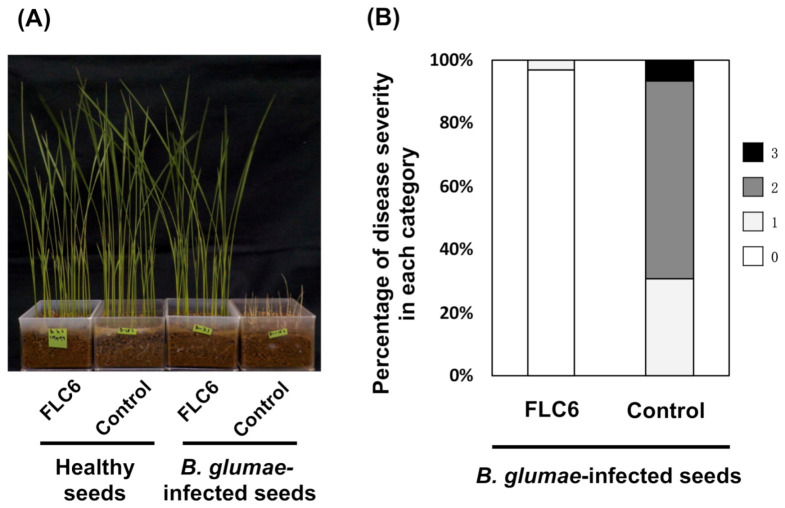
Disease suppression against rice seedling rot by FLC6 treatment. (**A**) Seedlings grown Figure 302746. with FLC6 treatment or without (control) at 8 days after sowing. (**B**) Proportion of rice seedling rot disease severity scores for *B. glumae* MAFF302746-infected seeds that were immersed in dilute FLC6 suspension or only a dilute potato-peptone-glucose (PPG) medium of the same strength as that used for preparing the FLC6 suspension as a control.

**Table 1 viruses-13-00591-t001:** Amino acid sequence similarity between FLC6 gp175 or gp176 and putative tail fiber proteins of RSF1 and RSL2.

Phage	NCBI RefSeq Accession	FLC6 gp175		FLC6 gp176	
		Sequence Identity ^1^ (%)	Query Coverage ^1^ (%)	Sequence Identity ^1^ (%)	Query Coverage ^1^ (%)
RSL2	YP_009213007.1	84	100	92	54
RSF1	YP_009207973.2	76	100	95	50

^1^ Sequence identity and query coverage were calculated by blastp search.

**Table 2 viruses-13-00591-t002:** Host range of FLC6.

Species	Strain	Sensitivity ^2^
*B. glumae*	MAFF106619	+
*B. glumae*	MAFF301169 ^1^	+
*B. glumae*	MAFF302417	+
*B. glumae*	MAFF302552	+
*B. glumae*	MAFF302746	+
*B. plantarii*	MAFF106727	+
*B. plantarii*	MAFF302466	+
*B. plantarii*	MAFF302475	+
*B. plantarii*	MAFF302909	+
*B. plantarii*	MAFF302912	+
*B. plantarii*	MAFF302936	+
*R. pseudosolanacearum*	MAFF106603	+
*R. pseudosolanacearum*	MAFF106611	+
*R. pseudosolanacearum*	MAFF211270	−
*R. pseudosolanacearum*	MAFF301485	−
*R. syzygii* subsp. *indonesiensis*	MAFF211271	−
*R. syzygii* subsp. *indonesiensis*	MAFF327032	−

^1^ MAFF301169 is a type strain of *B. glumae*.^2^ "+" means lysis of bacteria by FLC6, and "−" means no lysis by FLC6.

## Data Availability

The complete genome sequence of *Burkholderia* phage FLC6 is available at GenBank, EMBL, and DDBJ databases (accession number LC592711).

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
