# Peer review of "Isolation and Characterization of a Novel Jumbo Phage from Leaf Litter Compost and Its Suppressive Effect on Rice Seedling Rot Diseases"

_viruses, 2021, doi:10.3390/v13040591_

Round 1

Reviewer 1 Report

In this study, the authors isolated a new phage, Jumbo Burkholderia phage FLC6, from leaf litter compost. They observed the phage morphology using TEM and suggested that FLC6 belongs to the family Myoviridae in the order Caudovirales. They also determined the complete genomic DNA sequence and characterized it using bioinformatic tools. They analyzed the host range of FLC6 and demonstrated the disease suppression against rice seedling rot by FLC6 treatment. FLC6 has broad host range and considerable potential for use as a biocontrol agent against phytopathogenic bacteria in crop cultivation. Overall, this study was done well and the manuscript is clear. I just have some comments for the further improvement.

1). the authors showed intact particle in figure 1A. It looks fussy. How do you determine that? Not water contamination? Any controls?

2). the authors examined the susceptibility of four strains of R. pseudosolanacearum and two strains of R. syzygii subsp. indonesiensis to FLC6 based on sensitivity or no sensitivity. Any analysis to quantify or compare? Such as plaque size.

3). FLC6 has high genome/protein sequence similarity with RSL2 and RSF1, and also has some unique ORFs. It would be very helpful to compare their host ranges.

4). Regarding Figure 4. Phylogenetic tree, please provide higher resolution and bigger font size.

5). Please use g instead of rpm. Line 138

Author Response

Responses to the comments of Reviewer 1

Reviewer 1:

1) Reviewer’s comment 1:

The authors showed intact particle in figure 1A. It looks fussy. How do you determine that? Not water contamination? Any controls?

>Response: We thought your comment is about an intact particle of FLC6, and we replaced figure 1B with a clearer picture. FLC6 was purified by single plaque picking procedure. Using TEM, we could observe only 2 types of phage particles (with an intact tail like Figure 1B and with a contracted tail like Figure 1C). Since tail contraction is a common feature for Myoviridae, we decided both types as FLC6 particles.

“PPG liquid medium” for FLC6 culture, “Phage buffer” used to suspend FLC6 pellet after ultracentrifugation were prepared using distilled sterilized water, and all laboratory instruments were autoclaved. Therefore, we are considering the possibility of phage contamination is excluded.

We are afraid that we may misunderstood your comment. In case you indicated plaques shown in figure 1A, we replaced figure 1A with higher contrast. When we performed plaque assay, we made a plate using “Phage buffer” instead of phage solution as a negative control and confirmed there was no plaque.

2) Reviewer’s comment 2:

The authors examined the susceptibility of four strains of R. pseudosolanacearum and two strains of R. syzygii subsp. indonesiensis to FLC6 based on sensitivity or no sensitivity. Any analysis to quantify or compare? Such as plaque size.

>Response: We are sorry to not be able to quantify or compare the susceptibility of host bacteria, e.g., plaque size. Actually, plaque size is generally affected by some factors, e.g., bacterial growth speed, the length of latent period for phage multiplication in a bacterial host, and phage diffusion speed through top agar medium. Since the growth speed of bacterial hosts: four strains of R. pseudosolanacearum and two strains of R. syzygii subsp. indonesiensis was not same, we thought that it seems to be impossible to directly compare their plaque size among them.

3) Reviewer’s comment 3:

FLC6 has high genome/protein sequence similarity with RSL2 and RSF1, and also has some unique ORFs. It would be very helpful to compare their host ranges.

>Response: According to the suggestion, we added the sentences (line 355) as follows: “The host range of FLC6 to Ralstonia spp. was different from the reported host range of RSF1 and RSL2 (Supplementary Table S2).”

We also added Supplementary Table S2, which shows “Comparison of host range among FLC6, RSF1, and RSL2.”

4) Reviewer’s comment 4:

Regarding Figure 4. Phylogenetic tree, please provide higher resolution and bigger font size.

>Response: Figure 4 was enlarged and we replaced labels for bigger font size.

5) Reviewer’s comment 5:

Please use g instead of rpm. Line 138

>Response: We revised the “45,000 rpm” to “169,800 × g”.

Reviewer 2 Report

The study characterized phages leaf litter compost, which were screened for their potential to suppress rice seedling rot disease caused by the bacterium Burkholderia glumae. The study found a novel phage in the suspension of leaf litter compost. The study included the structural, genomic and host range characterization of these phages. A number of these phages showed a wide host range, which may make the phages especially advantageous as biocontrol agents against multiple bacterial diseases.

The procedure for isolating the phages and the results are well-described. A couple of details are needed:

Section 2.4. Before describing the sequencing protocol, were the phages purified (e.g. using cesium chloride gradient ultracentrifugation) to ensure that when sequencing and assembling the phage reads no host-related sequences were present before read assembly. This is particularly important as some phage and bacterial host sequences may be similar and may affect further assembly and annotation.

Was any control included in the host range analysis to ensure that there was no cross contamination with the bacterial hosts?

Author Response

Responses to the comments of Reviewer 2

Reviewer 2:

Reviewer’s comment 1:

Section 2.4. Before describing the sequencing protocol, were the phages purified (e.g. using cesium chloride gradient ultracentrifugation) to ensure that when sequencing and assembling the phage reads no host-related sequences were present before read assembly. This is particularly important as some phage and bacterial host sequences may be similar and may affect further assembly and annotation.

>Response: For DNA extraction, we first filtered phage liquid culture through a 0.45 µm pore-size membrane filter to remove bacterial cells. Then, the filtrate was ultracentrifuged to precipitate FLC6 particles. After re-suspending the pellet with phage buffer, phage DNA was extracted.

     We inserted the sentence how to prepare bacteria-free phage solution, before describing the sequencing protocol in revised text.

Reviewer’s comment 2:

Was any control included in the host range analysis to ensure that there was no cross contamination with the bacterial hosts?

>Response: Since we obtained their original stocks from NARO GeneBank for the bacterial hosts used in this study, bacterial strains used in this study are guaranteed. Throughout the experiments, we treated them carefully to avoid contamination. Thus, we are thinking that it is impossible to occur cross contamination with the bacterial hosts. Indeed, we could not find appropriate control in our experiments.

Round 2

Reviewer 1 Report

All my cooments are addressed. Thanks!

Author Response

Manuscript ID: viruses-1156300

Dear Prof. Dr. Elena G. Biosca and Dr. María Belén Álvarez Ortega,

Cc: Ms. Narcisa Fabian,

Thank you very much for your great effort to further improve our manuscript (Manuscript ID: viruses-1156300) entitled “Isolation and Characterization of a Novel Jumbo Phage from Leaf Litter Compost and Its Suppressive Effect on Rice Seedling Rot Diseases”.

We would like to thank editors and reviewers for giving us many valuable suggestions. We have marked all the revisions in the manuscript using Track Changes function in Microsoft Word for convenience. Our point-by-point revisions in the manuscript are shown on the following pages.

It would be greatly appreciated if you could consider our manuscript for publication in “Viruses”.

Yours sincerely,

Hideki Takahashi

E-mail: hideki.takahashi.d5@tohoku.ac.jp

For Editors

- Replace "phage solution" with "phage suspension" throughout the manuscript.

>Response: According to the suggestion, we replaced “phage solution” with “phage suspension” and also “FLC6 solution” with “FLC6 suspension.”

- Line 182. Replace "which has" with "at".

>Response: As you could point out, we replaced the words.

- Lines 337-338. Erwinia and Pantoea belong to the Erwinaceae family, please see: M. Adeolu, S. Alnajar, S. Naushad, S.G.R. Genome-based phylogeny and taxonomy of the ‘Enterobacteriales’: proposal for Enterobacterales ord. nov. divided into the families Enterobacteriaceae, Erwiniaceae fam. nov., Pectobacteriaceae fam. nov., Yersiniaceae fam. nov., Hafniaceae fam. nov., Morganellaceae fam. nov., and Budviciaceae fam. nov. Int. J. Syst. Evol. Microbiol. 66 (2016), pp. 5575-5599. 

>Response: We are so sorry that we wrote the wrong family name. We replaced “Enterobacteriaceae” with “Erwiniaceae” at line 365.

- Check the bibliographic citations to remove capital letters from the titles of the articles, for example citations 10, 13, 21, 23, 24, etc, and any missing italics, such as in the citation 15. 

>Response: From the title of the articles (citation 10, 13, 22-24, 38-40) and the title of the book chapters (citation 1, 4, 21, 29), we removed capital letters. According to your suggestion, we italicized the word “Myoviridae” in citation 15.

- Since Ralstonia phages are mentioned in the manuscript, some recent article/review on the phages of Ralstonia species with biocontrol activity from the last years are missing in the discussion. 

>Response: In this revision, we added the following sentence about the recent report about phage therapy against Ralstonia, at line 386. “So far, some jumbo Ralstonia phages were shown to have disease suppressing effect against bacterial wilt of tomato [8,42-44].” Along with this change, we added citation 42 and 43 to the reference section.
